# Association between COVID-19 vaccination and atopic diseases in US adults: A retrospective cohort study

Min Lu[1], Zixuan Bu[2], Nana Xiang[3], Juebo Yu[1]*

1 Department of Otolaryngology, Affiliated Hospital of Yangzhou University, Yangzhou University, Yangzhou, Jiangsu, China, 2 Department of clinical medicine, Medical College of Yangzhou University, Yangzhou, Jiangsu, China, 3 Department of Otolaryngology, Kunshan Hospital Integrated Traditional Chinese and Western Medicine, Kunshan, Suzhou, Jiangsu, China

* yujuebo2004@163.com

## Abstract

COVID-19 Vaccinations are associated with higher allergic reactions risk among adults. However, evidence on whether no vaccinated with COVID-19 vaccine is associated with fewer incidence among individuals with atopic diseases remains limited. This study is to investigate whether COVID-19 Vaccination is associated with increased risk of adult atopic diseases. A cross-sectional survey was conducted using data from the 2021 US National Health Interview Survey (NHIS) that included 29201 respondents aged 18 years or older adults. Multivariable logistic regression was conducted to estimate the association of COVID-19 vaccination and atopic disease. Crude and adjusted odds ratios (aORs) and 95% CIs were estimated. Analysis of the data was performed from October 01, 2023, to January 22, 2024. Of 29201 respondents (mean [SD] age, 52.6 [18.4] years; 13240 [45.3%] male), the US prevalence was 49.6% (unweighted, 95% CI, 49.1%−50.2%) from all years of the 2021 NHIS for self-reported hay fever, 13.7% (unweighted, 95% CI, 13.3%− 14.1%) for asthma, 10.9% (unweighted, 95% CI, 10.1%−11.3%) for skin allergy, 10.0% (unweighted, 95% CI, 9.7%−10.4%) for food allergy, and 45.1% (unweighted, 95% CI, 45.6%−45.7%) for no COVID-19 vaccination, 6.4% (95% CI, 6.1%−6.9%) for one COVID-19 vaccination, 43.1% (unweighted, 95% CI, 42.6%−43.7%) for two COVID-19 vaccinations, 5.3% (unweighted, 95% CI, 5.1%−5.6%) for more than 2 COVID-19 vaccinations. In multivariable analysis across the 2021 NHIS, COVID-19 vaccinations does not increase the risk of skin allergy (aOR, 1.03; 95%CI, 0.86–1.28; P = 0.135), asthma (aOR, 1.05; 95%CI, 0.98–1.13; P = 0.164), and food allergy (aOR, 1.03; 95%CI, 0.95–1.12; P = 0.437) in adults, compared with adults without COVID-19 vaccination; whereas, in patients with COVID-19 vaccination had significantly higher odds of hay fever (aOR, 1.21; 95% CI, 1.15–1.27; P < 0.001) compared with adults without COVID-19 vaccination. Further sensitivity and subgroup analysis suggested that the number of COVID-19 vaccinations was associated with newly diagnosed hay fever

**Data availability statement:** All relevant data for this study are publicly available from the Dryad repository (https://doi.org/10.5061/dryad.05qfttffr).

**Funding:** The author(s) received no specific funding for this work.

**Competing interests:** The authors have declared that no competing interests exist.

(aOR, 1.17 95% CI,1.10–1.24; P < 0.001) and asthma (aOR, 1.12 95% CI,1.03–1.27; P = 0.011). In this cross-sectional study, COVID-19 vaccination was associated with higher risk incidence of hay fever, but COVID-19 vaccination was not associated with an increased in incidence risk in skin allergy, asthma and food allergy. The determinants of the association of COVID-19 vaccination with atopic diseases should be identified in the future studies are needed.

## Introduction

Coronavirus disease 2019 (COVID-19) is caused by the novel coronavirus, Severe Acute Respiratory Syndrome Coronavirus 2 (SARS-CoV-2), which spread rapidly in 2020, causing a global pandemic with devastating medical and socioeconomic consequences. Developing safe and effective vaccinations quickly is one of the most important ways to contain a pandemic. SARS-CoV-2 vaccinations were developed quickly, which was a turning point in the pandemic's evolution and resulted in the saving of tens of millions of lives worldwide [1]. Adverse effects may be linked to vaccinations. Adverse occurrences may manifest as systemic or localized [2]. The reporting of allergic reactions, particularly at the beginning of SARS-CoV-2 vaccination with mRNA vaccines, the anaphylaxis and non-anaphylaxis allergic reactions based on spontaneous reports to VAERS, and increased public and health care provider awareness have become so significant for community acceptance of COVID-19 vaccines, even though local and systemic adverse events are the most frequently reported adverse events following SARS-CoV-2 vaccinations [3].

Allergic reactions to COVID-19 vaccines are thought to be driven by the excipients. Polyethylene glycol in mRNA vaccines and polysorbate 80 in viral vector-based vaccines are suspected as possible culprits, although this has not yet been proven. [4] Non–immunogloblin E immunologic mechanisms, possibly complement activation-related pseudoallergy, are thought to play a role in allergic reactions to polyethylene glycol [4, 5]. Immunoglobulin E (IgE)-mediated allergic diseases (hereon called allergies) is associated with chronic systemic inflammation and have a high and increasing prevalence in the general population, and also result in high morbidity and high costs for healthcare systems [6]. The most commonly reported allergies are atopic diseases (hay fever, asthma, eczema, and food allergies); and they have been associated with multiple comorbid chronic health conditions, such as rheumatoid arthritis, cardiovascular disease, type 1 diabetes, inflammatory bowel disease, anxiety or depression, impaired sleep, and multiple other mental comorbidities [7–12]. The incidence of atopic diseases to COVID-19 vaccines in the general population increased during the COVID-19 pandemic, [13, 14] but studies have not discriminated between adults with and without COVID-19 vaccination. Understanding the connection between adult atopic diseases and the COVID-19 vaccination could give public health care professionals low-cost, high-impact intervention strategies that could help adults live longer and better lives while also lowering adult morbidity and socioeconomic costs. We analyzed a 2021 US NHIS data set that included

self-reported information of COVID-19 and atopic diseases to determine whether there was an association between COVID-19 vaccination and atopic diseases in US adults.

## Methods

### Study sources

Deidentified data were assessed from the 2021 US National Health Interview Survey (NHIS) adults' health surveys. This is a cross-sectional survey implemented by the National Center for Health Statistics of the Centers for Disease Control and Prevention. The general US population's health status, healthcare services, lifestyle risk factors, and other health issues were collected using questionnaire-based in-person interviews with adult representatives of households. which included COVID-19 vaccination and atopic diseases and information on covariates. Participants with symptoms of atopic diseases will be included after the COVID-19 vaccine is administered. This study was based on secondary analyses of publicly available and deidentified NHIS data and was determined to be non–human participant research that did not require the Affiliated Hospital of Yangzhou University institutional review board review or informed consent.

### Ascertainment of main outcome

The main outcome of our interest was self-reported COVID-19 vaccination and atopic diseases, assessed by some questions. Doctors diagnose atopic diseases according to ICD-10 codes (L20(all sub-codes), J30 (all sub-codes),J45 (all sub-codes),and T78.0). The questions used in this study to assess for history of atopic disease and COVID-19 vaccination in adults (NHIS) are presented in Table 1. A schematic illustration of the participant selection process is presented in Fig 1.

### Self-reported covariates

Self-reported sociodemographic characteristics included age, sex (female and male), race/ ethnicity (non-Hispanic White, non-Hispanic Black, Hispanic, and other), marital status was recorded as married, widowed, divorced or separated, or never married or living with partner, educational level (identified as less than high school, high school, and above high

**Table 1. Questions used in our study.**

| Variable | Question text | Description | Respondents' answers |
|---|---|---|---|
| **Hay fever** | Do you get symptoms such as sneezing, runny nose, or itchy or watery eyes due to hay fever, seasonal or year-round allergies? | Current respiratory allergy | 1.yes; 2.no |
| **Hay fever** | Have you ever been told by a doctor or other health professional that you had hay fever, seasonal or year-round allergies? | Respiratory allergy diagnosis | 1.yes; 2.no |
| **Food allergy** | Do you have an allergy to one or more foods? | Current food allergy | 1.yes; 2.no |
| **Food allergy** | Have you ever been told by a doctor or other health professional that you had an allergy to one or more foods? | Food allergy diagnosis | 1.yes; 2.no |
| **Skin allergy** | Have you ever been told by a doctor or other health professional that you had eczema or atopic dermatitis? | Skin allergy diagnosis | 1.yes; 2.no |
| **Skin allergy** | The next question is about an allergic skin condition. Do you get an itchy rash due to eczema or atopic dermatitis? | Current skin allergy | 1.yes; 2.no |
| **Asthma** | Have you EVER been told by a doctor or other health professional that you had asthma? | Ever had asthma | 1.yes; 2.no |
| **Asthma** | Do you still have asthma? | Current asthma | 1.yes; 2.no |
| **COVID-19 Vaccination** | Have you had a COVID-19 vaccination? | COVID-19 vaccination | 1.yes; 2.no |
| **Vaccine dose** | How many COVID-19 vaccinations have you received? | Number of COVID-19 vaccinations | 1. Vaccination 2. two vaccinations 3. more than 2 vaccinations. |

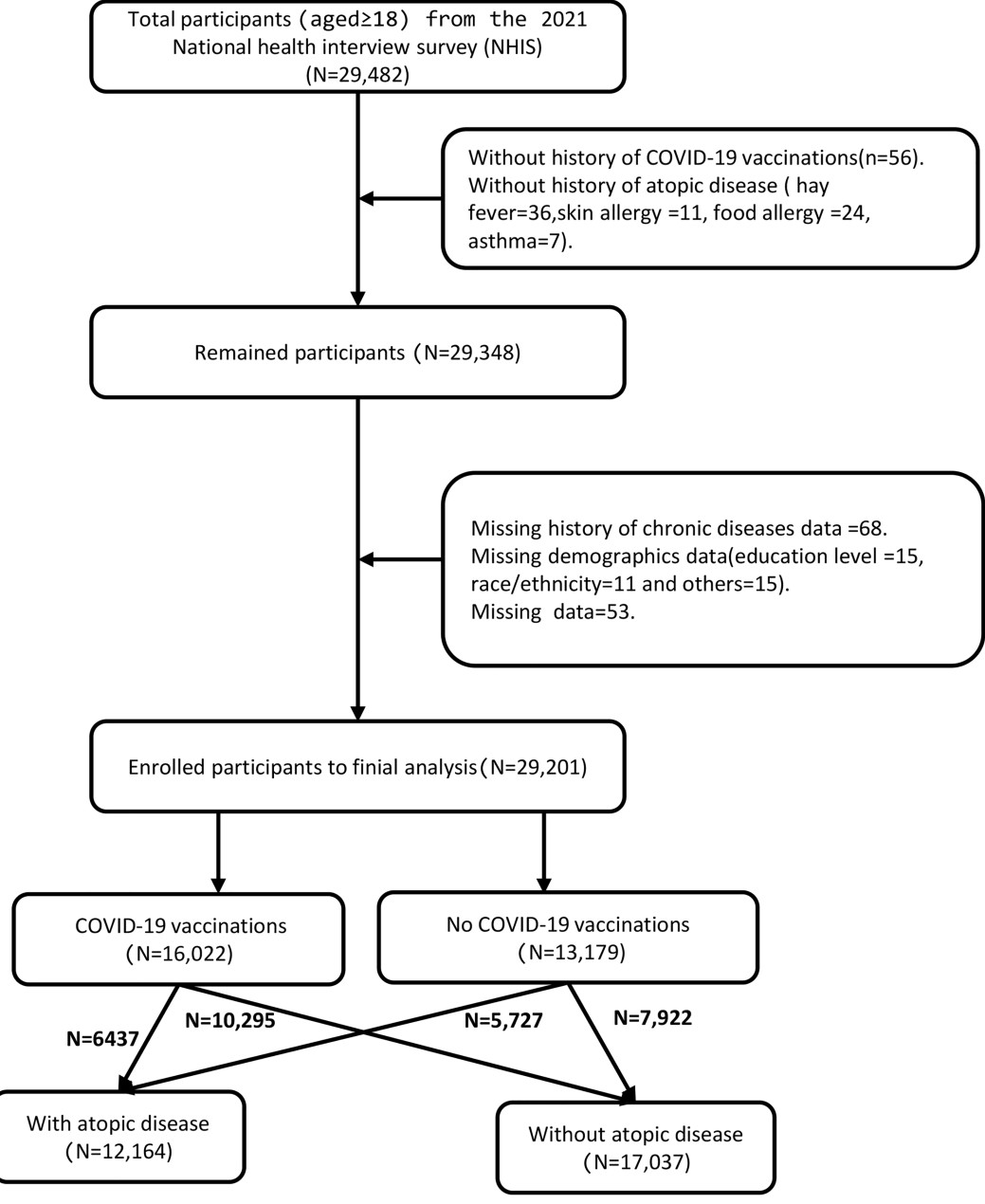

**Fig 1. A schematic illustration of the participant selection process.**

school), and family income(ratio of family income to the federal poverty level was categorized as <1.30, 1.30–3.49, or ≥3.5). Lifestyle risk factors included smoking status (never, former, current, or missing or unknown), Body mass index [BMI; calculated as weight in kilograms divided by height in meters squared) was calculated using self-reported weight and height (underweight, < 18.5; normal weight, 18.5–24.9; overweight, 25.0–29.9; and obese ≥30.0)]. The presence of the following medical diagnosis at any point in the respondent's lifetime were also obtained at baseline: hypertension, hypercholesterolemia, type 1 and 2 diabetes, Covid-19, cancer, depression, and sleep diseases, and coronary artery

disease (history of heart attack, history of myocardial infarction, history of angina, history of congestive heart failure, angioplasty, or cardiac surgery).

## Statistical analysis

Low rates of missing values were found in the 2021 NHIS data for the history of atopic illness, COVID-19 vaccination, and other variables. Complete data analysis was conducted for all variables; that is, individuals with missing data were not allowed to participate. When continuous variables had a normal distribution, they were given as mean (SD) and, when they didn't, as median (interquartile range [IQR]). In the case of categorical variables, frequency (%) was recorded. In univariate analysis, they were compared using Mann-Whitney U tests and χ2/Fisher exact tests, respectively.

The dependent variable was COVID-19 vaccination and the independent variable was history of atopic diseases (yes or no). Multiple logistic regression was used to model adjusted proportions of categorical dependent variables. Most estimates of regression adjusted for baselines covariates, including age (continuous), sex (male or female), race/ethnicity(Hispanic, Non-Hispanic White only, Non-Hispanic Black/African American only, Non-Hispanic Asian only, Non-Hispanic other races and other/M-ethnicity), education level(less than high school, High school graduate or GED, and More than high school), cigarette smoking status(nonsmokers, former smokers, and current smokers), urban–rural classification(large central metro, large fringe metro, medium and small metro, and nonmetropolitan), current marital status(married divorced/separated/widowed, never married, living with a partner, and unknown marital status), region(northeast, midwest, south, and west), and family income-to-poverty ratio( < 1, 1.0–2.0, 2.0–3.9, and > 4),BMI categories(underweight, healthy weight, overweight, obese, and unknown), and chronic comorbidities[ever had COVID-19, history of diabetes, history of hypertension, history of CAD, history of COPD, history of high cholesterol level, history of cancer, History of anxiety, history of depression, and history of weakened immune system(due to prescriptions or health condition)].

A subgroup analyses was further stratified by age (18–44, 45–60, 61–74,75–84, ≥ 85), sex (male or female), race/ethnicity(Hispanic, Non-Hispanic White only, Non-Hispanic Black/African American only, Non-Hispanic Asian only, Non-Hispanic other races and other/M-ethnicity), education level(less than high school, High school graduate or GED, and More than high school), cigarette smoking status(nonsmokers, former smokers, and current smokers), urban–rural classification(large central metro, large fringe metro, medium and small metro, and nonmetropolitan),current marital status(married divorced/separated/widowed, never married, living with a partner, and unknown marital status), region(northeast, midwest, south, and west), and family income-to-poverty ratio(<1, 1.0–2.0, 2.0–3.9, and ≥4), BMI categories(underweight, healthy weight, overweight, obese, and unknown), and chronic comorbidities [yes/no; including ever had COVID-19, history of diabetes, history of hypertension, history of CAD, history of COPD, history of high cholesterol level, history of cancer, History of anxiety, history of depression, and history of weakened immune system(due to prescriptions or health condition)]. The P values for the production terms between atopic diseases and COVID-19 vaccination and the stratified factors were used to estimate the significance of interactions.

We conducted several sensitivity analyses. First, because one of our goals was to evaluate the association of COVID-19 vaccination with atopic diseases, we estimated the association of COVID-19 vaccination at baseline (no at baseline, 1 vaccination, 2 vaccinations, and more 2 vaccinations) with COVID-19 vaccination dose among participants in the with or without atopic disease group. Second, to minimize the potential reverse causation bias, we excluded participants with a history of atopic diseases were further excluded from the main analyses. we estimated our model using newly diagnosed atopic diseases as the outcome. Newly diagnosed atopic diseases are equal to the current diagnosis minus prior history of atopic disease.

Estimates with crude and adjusted odds ratios(aORs) and 95% confidence intervals (CIs) that did not include the null or P value less than.05 were considered statistically significant with 2-sided tests. All analyses were performed using IBM SPSS 24 (IBM, Armonk, NY, USA) statistical software.

## Results

### Baseline characteristics

Participants In the 2021 NHIS cycles included in the study, there were 29482 respondents aged 18 years or older. After removing individuals missing or invalid data on COVID-19 vaccination and atopic diseases in the past year, the cohort consisted of 29201 respondents; demographic and clinical data are displayed in the Table 2. The mean (SD) age was 52.6 (18.4) years; 15961 respondents (54.7%) were female, and 13240 (45.3%) were male. Aged people, female, white, medium and small metropolitan, nonsmokers, higher education level, south region people, overweight, higher family income, have History of diabetes, history of CVD, history of COPD, history of high cholesterol level, history of cancer and history of weakened immune system were more likely to receive the COVID-19 vaccination and multiple COVID-19 vaccinations (Table 2). The US prevalence was 49.6% (unweighted, 95% CI, 49.1%−50.2%) from all years of the 2021 NHIS for self-reported hay fever, 13.7% (unweighted,95% CI, 13.3%− 14.1%) for asthma, 10.9% (unweighted, 95% CI,10.1%−11.3%) for skin allergy, 10.0% (unweighted,95% CI, 9.7%−10.4%) for food allergy, and 45.1% (unweighted,95% CI, 45.6%−45.7%) for no COVID-19 vaccination, 6.4% (95% CI, 6.1%−6.9%) for one COVID-19 vaccination, 43.1% (unweighted, 95% CI, 42.6%−43.7%) for two COVID-19 vaccinations, 5.3% (unweighted, 95% CI, 5.1%−5.6%) for more than 2 COVID-19 vaccinations. (Table 3).

### Association between atopic disease and COVID-19 vaccinations

The association of atopic disease and COVID-19 vaccination indicated an unadjusted OR of 1.03(95% CI, 0.91−1.16; p = 0.692), 1.29 (95% CI, 1.20−1.39; p < 0.001), 1.03(95% CI, 0.96−1.10; p = 0.414) and 1.07 (95% CI, 0.95−1.22; p = 0.276) in patients with skin allergy, hay fever, asthma and food allergy, respectively. In multivariable analysis across the NHIS, COVID-19 vaccinations does not increase the risk of skin allergy (aOR, *1.03; 95%CI, 0.86−1.28; P = 0.135)*, asthma (aOR, 1.05;95%CI,0.98−1.13;P = 0.164), and food allergy(aOR, 1.03; 95%CI, 0.95−1.12; P = 0.437) in adults, compared with adults without COVID-19 vaccination; whereas, in patients with COVID-19 vaccination had significantly higher odds of hay fever (aOR, 1.21;95% CI, 1.15−1.27;P < .001) compared with adults without COVID-19 vaccination (Table 3).

Furthermore, analysis of covariates indicated no associations between the number of comorbid atopic disorders examined and COVID-19 vaccination. The overall association with a single atopic disorder was 0.98(95% CI, 0.76−1.27; p = 0.897),the association was not statistically significant in those with two atopic disorders (aOR,1.07 (95% CI, 0.82−1.38; p = 0.620), and no association were found (OR = 1.19 (95% CI, 0.91−1.54; p = 0.201) in patients with three atopic disorders and (OR = 1.20 (95% CI, 0.91−1.59; p = 0.201) in patients with four atopic disorders. Subsequent subgroup analysis of associations between COVID-19 vaccination and the risk of hay fever are shown in Table 4. The association between COVID-19 vaccination and hay fever in the stratified analysis was consistent with that in the multivariable logistic regression analysis. The stratified analysis demonstrated a statistically significant association between hay fever and COVID-19 vaccination in aged 40−60 years subjects (adjusted OR, 1.23; 95% CI, 1.11−1.36),Hispanic subjects(aOR, 1.32; 95% CI, 1.17−1.49), subjects with more than high school(aOR, 1.19; 95% CI, 1.12−1.27), subjects with former smoking habits (aOR, 1.20; 95% CI, 1.09−1.33), subjects living in Large central metro-metropolitan(aOR, 1.20; 95% CI, 1.10−1.31), and in west region(aOR, 1.38; 95% CI, 1.22−1.55), subjects with higher incomes (aOR, 1.30; 95% CI, 1.07−1.58), with hypertension (aOR, 1.20; 95% CI, 1.11−1.31),with hyperlipidemia (aOR, 1.20; 95% CI, 1.13−1.27), had history of cancer(aOR, 1.19; 95% CI, 1.03−1.24), with anxiety (aOR, 1.23; 95% CI, 1.08−1.39), and with depression (aOR, 1.24; 95% CI, 1.10−1.39), and with weakened immune system (aOR, 1.17; 95% CI, 1.14−1.23). The interaction analysis revealed interactive role in the association between COVID-19 vaccination and hay fever (Table 4).

### Sensitivity analyses

We further analyzed the relationship between the number of COVID-19 vaccinations and atopic disorders. The findings suggested that the number of COVID-19 vaccinations were not associated with risk of skin allergy, and food allergy, but

**Table 2. Demographics of respondents by COVID-19 vaccination and number of COVID-19 vaccinations.**

| Variable | Respondents a | | | | |
|---|---|---|---|---|---|
| | Total | No COVID-19 vaccination | COVID-19 vaccination | | |
| | | | 1 vaccination | 2 vaccinations | More than 2 vaccinations |
| **Unweighted sample** | 29201(100) | 13179(45.1) | 1869(6.4) | 12597(43.1) | 1556(5.3) |
| **Sex** | | | | | |
| Male | 13240(45.3) | 6044(45.9) | 886(47.4) | 5631(44.7) | 679(43.6) |
| Female | 15961(54.7) | 7135(54.1) | 983(52.6) | 6966(55.3) | 877(56.4) |
| **Age categories, y** | 52.6(18.4) | 49.4(18.3) | 49.6(17.4) | 55.1(17.9) | 62.9(16.6) |
| **Race/ethnicity** | | | | | |
| Hispanic | 4053(13.9) | 1783(13.5) | 281(15.0) | 1772(14.1) | 217(13.9) |
| Non-Hispanic White only | 19447(66.6) | 8745(66.4) | 1227(65.7) | 8427(66.9) | 1048(67.4) |
| Non-Hispanic Black/African American only | 3140(10.8) | 1461(11.1) | 184(9.8) | 1353(10.7) | 142(9.1) |
| Non-Hispanic Asian only | 1797(6.2) | 822(6.2) | 132(7.1) | 731(5.8) | 112(7.2) |
| Non-Hispanic other races | 401(1.4) | 210(1.6) | 19(1.0) | 153(1.2) | 19(1.2) |
| Other/M-ethnicity | 363(1.2) | 158(1.2) | 26(1.4) | 161(1.3) | 18(1.2) |
| **Education level** | | | | | |
| Less than high school | 2560(8.8) | 1383(10.5) | 175(9.4) | 931(7.4) | 71(4.6) |
| High school graduate or GED | 7215(24.7) | 3818(29.0) | 476(25.5) | 2647(21.0) | 274(17.6) |
| More than high school | 19426(66.5) | 7978(60.5) | 1218(65.2) | 9019(71.6) | 1211(77.8) |
| **Cigarette smoking status** | | | | | |
| Nonsmokers | 20982(71.9) | 9510(72.2) | 1384(74.1) | 9048(71.8) | 1040(66.8) |
| Former smokers | 7338(25.1) | 3017(22.9) | 446(23.9) | 3383(26.9) | 492(31.6) |
| Current smokers | 881(3.0) | 652(4.9) | 39(2.1) | 166(1.3) | 24(1.5) |
| **Urban–Rural classification** | | | | | |
| Large central metro | 8863(30.4) | 3702(28.1) | 572(30.6) | 4103(14.1) | 486(31.2) |
| Large fringe metro | 6891(23.6) | 2884(21.9) | 448(24.0) | 3169(25.2) | 390(25.1) |
| Medium and small metro | 9214(31.6) | 4326(32.8) | 563(30.1) | 3834(30.4) | 491(31.6) |
| Nonmetropolitan | 2920(14.5) | 2267(17.2) | 286(15.3) | 1491(11.8) | 189(12.1) |
| **Current marital status** | | | | | |
| Married | 13298(45.5) | 6001(45.5) | 849(45.4) | 5713(45.4) | 735(47.2) |
| Divorced/separated/widowed | 7097(24.3) | 3222(24.4) | 483(25.8) | 3029(24.0) | 363(23.3) |
| Never married | 5998(20.5) | 2686(20.4) | 380(20.3) | 2607(20.7) | 325(20.9) |
| Living with a partner | 1826(6.3) | 826(6.3) | 108(5.8) | 811(6.4) | 81(5.2) |
| Unknown marital status | 982(3.4) | 444(3.4) | 49(2.6) | 437(3.5) | 52(3.3) |
| **Region** | | | | | |
| Northeast | 4728(16.2) | 1902(14.4) | 332(17.8) | 2237(17.8) | 257(16.5) |
| Midwest | 6261(21.4) | 2890(21.9) | 369(19.7) | 2663(21.1) | 339(21.8) |
| South | 10631(36.4) | 5155(39.1) | 699(37.4) | 4273(33.9) | 504(32.4) |
| West | 7581(26.0) | 3232(24.5) | 469(25.1) | 3424(27.2) | 456(29.3) |
| **BMI categories** | | | | | |
| Underweight | 460(1.6) | 226(1.7) | 22(1.2) | 184(1.5) | 28(1.8) |
| Healthy weight | 9065(31.0) | 3999(30.3) | 586(31.4) | 3966(31.5) | 514(33.0) |
| Overweight | 9827(33.7) | 4416(33.5) | 628(33.6) | 4235(33.6) | 548(35.2) |
| Obese | 9127(31.3) | 4157(31.5) | 597(31.9) | 3950(31.4) | 423(27.2) |
| Unknown | 722(2.5) | 381(2.9) | 36(1.9) | 262(2.1) | 43(2.8) |
| **Family income-to-poverty ratio, mean(SD)** | 4.24(2.99) | 3.75(2.80) | 4.18(3.02) | 4.65(3.10) | 5.26(3.14) |

*(Continued)*

**Table 2.** (Continued)

| Variable | Respondents a | | | | |
|---|---|---|---|---|---|
| | Total | No COVID-19 vaccination | COVID-19 vaccination | | |
| | | | 1 vaccination | 2 vaccinations | More than 2 vaccinations |
| **Chronic comorbidities** | | | | | |
| Ever had COVID-19 | 3492(12.0) | 1620(12.3) | 227(12.1) | 1467(11.6) | 178(11.4) |
| History of diabetes | 3104(10.6) | 1219(9.2) | 214(11.4) | 1450(11.5) | 221(14.2) |
| History of hypertension | 10580(36.2) | 4273(32.4) | 607(32.5) | 4946(39.3) | 754(48.5) |
| History of CVD | 2989(10.2) | 1239(9.4) | 160(8.6) | 1364(10.8) | 226(14.5) |
| History of pulmonary diseases b | 1685(5.8) | 731(5.5) | 112(6.0) | 726(5.8) | 116(7.5) |
| History of high cholesterol level | 9008(30.8) | 3449(26.2) | 522(27.9) | 4340(34.5) | 697(44.8) |
| History of cancer | 3618(12.4) | 1321(10.0) | 179(9.6) | 1786(14.2) | 332(21.3) |
| History of anxiety | 4823(16.5) | 2206(16.7) | 331(17.7) | 2052(16.3) | 234(15.0) |
| History of depression | 5315(18.2) | 2379(18.1) | 348(18.6) | 2332(18.5) | 256(16.5) |
| History of weakened immune system c | 1252(4.3) | 520(3.9) | 85(4.5) | 538(4.3) | 109(7.0) |

Abbreviation: GED, General Educational Development; CVD, Cardiovascular Diseases;

a Data are presented as number (percentage) of respondents unless otherwise indicated.

b including Chronic Obstructive Pulmonary Disease, emphysema, or chronic bronchitis.

c due to prescriptions or health condition.

it was associated with an increased the risk of hay fever and asthma (Table 5). After adjustment for age, sex, race/ethnicity, education level, cigarette smoking status, urban–rural classification, current marital status, region, family income-to-poverty ratio, BMI, and chronic comorbidities, the multivariable-adjusted ORs for hay fever were 1.00 (reference) for no COVID-19 vaccination, 1.15 (95% CI,1.04–1.27) for 1 vaccination, 1.19 (95% CI,1.13–1.25) for 2 vaccinations, and 1.09 (95% CI,0.98–1.22) for more than 2 vaccinations (P<0.001 for trend). The multivariable-adjusted OR for the risk of asthma were 1.00 (reference) for no COVID-19 vaccination, 0.88 (95% CI, 0.76–1.03) for 1 vaccination, 1.04 (95% CI, 0.96–1.12) for 2 vaccinations, and 1.21 (95% CI, 1.03–1.42) for more than 2 vaccinations (P=0.015 for trend). Additionally, we investigated the relationship between COVID-19 vaccination and the newly diagnosed atopic disorders. The results showed that The multivariable-adjusted OR for the risk of still asthma(n=2,471) was 1.12 (95% CI,1.03–1.27;P=0.011), for the risk of newly diagnosed skin allergy (n=698; 2,496 of 3,230 total skin-allergy cases were excluded because of pre-existing disease)was 0.98 (95% CI,0.84–1.12;P=0.799), for the risk of newly diagnosed hay fever (7,728 of 14,491 total hay-fever cases were excluded because of pre-existing disease) was 1.17 (95% CI,1.10–1.24;P<0.001), and for the risk of newly diagnosed food allergy(n=1,084;1,838 of 2,922 total food-allergy cases were excluded because of pre-existing disease) was 1.01 (95% CI,0.89–1.15;p=0.823).

## Discussion

Findings from this nationally representative cohort of US adults support the idea that the people who received COVID-19 vaccinations are associated with greater self-reported current hay fever risk compared with those who did not receive COVID-19 vaccination. On the other hand, we did not find a consistent similar association of self-reported current skin allergy, asthma, and food allergy with COVID-19 vaccination. We found that the odds of self-reported current hay fever and asthma increased with the number of COVID-19 vaccinations present. In contrast, skin allergy and food allergy were not associated with the number of COVID-19 vaccinations. Furthermore, the odds of the number of atopic disorders did not increase with the number of COVID-19 vaccinations received. The results of sensitivity analysis showed that the

**Table 3. Association between adult atopic disease and COVID-19 vaccination in the NHIS, 2021.**

| | No COVID-19 vaccination | | COVID-19 vaccination [a] | | | | | |
|---|---|---|---|---|---|---|---|---|
| | Frequency | % Prevalence (95% CI) | Frequency% | % Prevalence (95% CI) | Crude OR (95% CI) | *P* Value | Adjusted OR(95% CI) | *P* Value |
| **Atopic Disease, No.** | | | | | | | | |
| **Total** | 13179 | | 16022 | | | | | |
| 0 | 5727 | 47.1(46.2-48.0) | 6437 | 53.0(52.0-53.8) | 1 [Reference] | | 1 [Reference] | |
| 1 | 4919 | 44.1(43.2-45.0) | 6236 | 55.9(55.0-56.8) | 1.10(0.85-1.41) | 0.469 | 0.98(0.76-1.27) | 0.897 |
| 2 | 1893 | 42.6(41.2-44.1) | 2546 | 57.4(55.9-58.8) | 1.24(0.96-1.59) | 0.096 | 1.07(0.82-1.38) | 0.620 |
| 3 | 517 | 43.3(40.5-46.1) | 677 | 56.7(53.9-59.5) | 1.31(1.02-1.70) | 0.037 | 1.19(0.91-1.54) | 0.201 |
| 4 | 123 | 49.4(43.2-55.6) | 126 | 50.6(44.3-46.8) | 1.29(0.97-1.68) | 0.079 | 1.20(0.91-1.59) | 0.201 |
| **Asthma** | | | | | | | | |
| No | 11348 | 45.0(44.4-45.7) | 13849 | 55.0(54.3-55.6) | 1 [Reference] | | 1 [Reference] | |
| Yes | 1831 | 45.7(44.2-47.27) | 2173 | 54.3(52.7-55.8) | 1.03(0.96-1.10) | 0.414 | 1.05(0.98-1.13) | 0.164 |
| **Skin allergy** | | | | | | | | |
| No | 11744 | 45.2(44.6-45.8) | 14263 | 54.8(54.3-55.4) | 1 [Reference] | | 1 [Reference] | |
| Yes | 1435 | 44.9(44.6-45.8) | 1759 | 56.2(54.5-57.9) | 1.03(0.91-1.16) | 0.692 | 1.03(0.86-1.28) | 0.135 |
| **Hay fever** | | | | | | | | |
| No | 7003 | 47.6(46.8-48.4) | 7707 | 52.4(51.6-53.2) | 1 [Reference] | | 1 [Reference] | |
| Yes | 6176 | 42.6(41.8-43.4) | 8315 | 57.4(56.6-58.2) | 1.29(1.20-1.39) | <0.001 | 1.21(1.15-1.27) | <0.001 |
| **Food allergy** | | | | | | | | |
| No | 11873 | 45.2(44.6-45.8) | 14406 | 54.8(54.2-55.4) | 1 [Reference] | | 1 [Reference] | |
| Yes | 1306 | 43.6(41.9-45.4) | 1616 | 54.0(52.2-55.8) | 1.07(0.95-1.22) | 0.276 | 1.03(0.95-1.12) | 0.437 |

Abbreviations: NHIS, National Health Interview Survey; OR, odds ratio.

a Binary logistic regression analyses were conducted with COVID-19 vaccination status as the binary outcome variable and the presence of asthma, skin allergies, hay fever, food allergies, and the count of atopic disorders as the binary predictor variables. We performed multivariable-adjusted models that accounted for various covariates, including age (continuous), sex (male or female), race/ethnicity (Hispanic, Non-Hispanic White only, Non-Hispanic Black/African American only, Non-Hispanic Asian only, Non-Hispanic other races and other/multi-ethnicity), education level (less than high school, high school graduate or GED, and more than high school), cigarette smoking status (nonsmokers, former smokers, and current smokers), urban–rural classification (large central metro, large fringe metro, medium and small metro, and non-metropolitan), current marital status (married, divorced/separated/widowed, never married, living with a partner, and unknown marital status), region (northeast, midwest, south, and west), and family income-to-poverty ratio (<1, 1.0–2.0, 2.0–3.9, and ≥4), BMI categories (underweight, healthy weight, overweight, obese, and unknown), and chronic comorbidities [yes or no; history of COVID-19, diabetes, hypertension, CVD, chronic obstructive pulmonary disease (COPD), high cholesterol level, cancer, anxiety, depression, and weakened immune system (due to prescriptions or health condition)]. Additionally, the multivariable models for each atopic disease included the other atopic diseases as covariates. We calculated crude and adjusted odds ratios (ORs) with their corresponding 95% confidence intervals (CIs).

vaccination of COVID-19 vaccine was related to the increase of incidence rate of newly diagnosed hay fever and still asthma.

The results of our study suggested that there was no direct correlation between COVID-19 vaccination and the increased risk of asthma, food allergy and skin allergy, but there was a correlation between the increased risk of hay fever and COVID-19 vaccination. Therefore, we conducted a subgroup analysis of COVID-19 vaccination and hay fever. Subgroup analysis showed that individuals with aged 40−60 years, more than high school, former smoking habits, living in Large central metro-metropolitan and in west region and higher incomes, hypertension, hyperlipidemia, anxiety, depression or weakened immune system more prone to hay fever than no COVID-19 vaccination group, the association remained significant after adjustment for these risk factors. Moreover, we found interactive role in the association between COVID-19 vaccination and hay fever, suggesting that COVID-19 vaccination was a risk factor associated with the incidence of hay fever.

As previously reported, Atopic disease has been shown to be associated with several different multiple comorbid chronic health conditions, including sleep disturbances [11, 12], being overweight and obese [15], infections [16], anxiety

**Table 4. Subgroups analysis association of hay fever with COVID-19 stratified by participant characteristics.**

| Subgroups | No COVID-19 vaccination | | COVID-19 vaccination | | aOR(95% CI) a | P for interaction |
|---|---|---|---|---|---|---|
| | Without hay fever | With hay fever | Without hay fever | With hay fever | | |
| **Sex** | | | | | | 0.118 |
| Male | 3535 | 2509 | 3787 | 2409 | 1.22(1.13-1.31) | |
| Female | 3468 | 3667 | 3920 | 4906 | 1.15(1.08-1.23) | |
| Age groups, years | | | | | | <0.001 |
| 18-44 | 3136 | 2700 | 2369 | 2623 | 1.17(1.08-1.27) | |
| 45-60 | 1706 | 1588 | 1853 | 2164 | 1.23(1.11-1.36) | |
| 61-74 | 1365 | 1289 | 2171 | 2359 | 1.14(1.03-1.26) | |
| 75-84 | 550 | 440 | 944 | 874 | 1.18(1.00-1.40) | |
| ≥85 | 246 | 159 | 370 | 295 | 1.24(0.95-1.62) | |
| **Race/Ethnicity** | | | | | | <0.001 |
| Hispanic | 982 | 801 | 1102 | 1168 | 1.32(1.17-1.49) | |
| Non-Hispanic White only | 4614 | 4131 | 5157 | 5545 | 1.21(1.13-1.28) | |
| Non-Hispanic Black/African American only | 774 | 687 | 779 | 900 | 1.28(1.11-1.48) | |
| Non-Hispanic Asian only | 437 | 385 | 484 | 491 | 1.16(0.95-1.40) | |
| Non-Hispanic other races | 114 | 96 | 95 | 96 | 1.21(0.82-1.79) | |
| Other/Ethnicity | 82 | 76 | 90 | 115 | 1.36(0.90-2.08) | |
| **Education** | | | | | | <0.001 |
| Less than high school | 865 | 518 | 705 | 472 | 1.14(0.96-1.36) | |
| High school graduate or GED | 2158 | 1660 | 1784 | 1613 | 1.14(1.05-1.28) | |
| More than high school | 3980 | 3998 | 5218 | 6230 | 1.19(1.12-1.27) | |
| **Cigarette smoking status** | | | | | | 0.008 |
| Nonsmokers | 5128 | 4382 | 5628 | 5844 | 1.15(1.09-1.22) | |
| Former smokers | 1475 | 1542 | 1943 | 2378 | 1.20(1.09-1.33) | |
| Current smokers | 400 | 252 | 136 | 93 | 1.07(0.77-1.47) | |
| **Urban -Rural** | | | | | | <0.001 |
| Large central metro | 2143 | 1559 | 2658 | 2503 | 1.20(1.10-1.31) | |
| Large fringe metro | 1525 | 1359 | 1937 | 2070 | 1.16(1.04-1.28) | |
| Medium and small metro | 2237 | 2089 | 2236 | 2652 | 1.19(1.10-1.30) | |
| Nonmetropolitan | 1098 | 1169 | 876 | 1090 | 1.12(0.99-1.28) | |
| **Current marital status** | | | | | | 0.121 |
| Married | 3225 | 2776 | 3532 | 3765 | 1.19(1.11-1.28) | |
| Divorced/separated/widowed | 1699 | 1523 | 1870 | 2005 | 1.17(1.06-1.29) | |
| Never married | 1407 | 1279 | 1606 | 1706 | 1.14(1.02-1.27) | |
| Living with a partner | 441 | 385 | 447 | 553 | 1.36(1.11-1.66) | |
| Unknown marital status | 231 | 213 | 252 | 286 | 1.14(0.86-1.50) | |
| **Region** | | | | | | <0.001 |
| West | 1079 | 823 | 1382 | 1444 | 1.38(1.22-1.55) | |
| Midwest | 1505 | 1385 | 1606 | 1765 | 1.15(1.37-1.28) | |
| Northeast | 2635 | 2520 | 2544 | 2932 | 1.16(1.07-1.25) | |
| South | 1784 | 147 | 2175 | 2174 | 1.23(1.11-1.34) | |

*(Continued)*

**Table 4.** (Continued)

| Subgroups | No COVID-19 vaccination | | COVID-19 vaccination | | aOR(95% CI) a | P for interaction |
|---|---|---|---|---|---|---|
| | Without hay fever | With hay fever | Without hay fever | With hay fever | | |
| **BMI categories** | | | | | | 0.430 |
| Underweight | 142 | 84 | 138 | 96 | 1.13(0.74-1.72) | |
| Healthy weight | 2231 | 1768 | 2596 | 2470 | 1.20(1.10-1.31) | |
| Overweight | 2376 | 2040 | 2624 | 2782 | 1.21(1.16-1.32) | |
| Obese | 2028 | 2129 | 2186 | 2784 | 1.16(1.06-1.27) | |
| Unknown | | | | | | |
| **Family income-to-poverty ratio** | | | | | | <0.001 |
| <1 | 914 | 749 | 617 | 584 | 1.28(0.84-1.96) | |
| 1.0-1.9 | 1513 | 1191 | 1257 | 1120 | 0.87(0.63-1.21) | |
| 2.0-3.9 | 2073 | 1886 | 2196 | 2291 | 1.13(0.89-1.44) | |
| ≥4 | 2503 | 2350 | 3637 | 4320 | 1.30(1.07-1.58) | |
| **Chronic conditions** | | | | | | |
| **Ever had COVID-19** | | | | | | 0.969 |
| No | 6063 | 5496 | 6664 | 7486 | 1.14(0.95-1.36) | |
| Yes | 940 | 680 | 1043 | 829 | 1.01(0.61-1.66) | |
| **History of diabetes** | | | | | | 0.529 |
| No | 6417 | 5543 | 6813 | 7324 | 1.12(0.94-1.34) | |
| Yes | 586 | 633 | 894 | 991 | 0.82(0.42-1.60) | |
| **History of hypertension** | | | | | | <0.001 |
| No | 4883 | 4023 | 4848 | 4867 | 1.18(1.11-1.26) | |
| Yes | 2120 | 2153 | 2859 | 3448 | 1.20(1.11-1.31) | |
| **History of CVD** | | | | | | 0.280 |
| No | 6402 | 5538 | 6879 | 7393 | 1.15(1.09-1.22) | |
| Yes | 601 | 638 | 828 | 922 | 0.93(0.73-1.18) | |
| **History of high cholesterol level** | | | | | | <0.001 |
| No | 5398 | 4332 | 5273 | 5190 | 1.20(1.13-1.27) | |
| Yes | 1605 | 1884 | 2434 | 3125 | 1.13(1.03-1.24) | |
| **History of cancer** | | | | | | 0.002 |
| No | 6368 | 5490 | 6682 | 7043 | 1.17(1.11-1.24) | |
| Yes | 635 | 686 | 1025 | 1272 | 1.19(1.03-1.37) | |
| **History of anxiety** | | | | | | 0.003 |
| No | 6169 | 4804 | 6851 | 6554 | 1.16(1.10-1.23) | |
| Yes | 834 | 1372 | 856 | 1761 | 1.23(1.08-1.39) | |
| **History of depression** | | | | | | <0.001 |
| No | 6072 | 4728 | 6714 | 6372 | 1.16(1.10-1.22) | |
| Yes | 931 | 1448 | 993 | 1943 | 1.24(1.10-1.39) | |
| **Weakened immune system b** | | | | | | 0.340 |
| No | 6767 | 5892 | 7415 | 7875 | 1.17(1.14-1.23) | |
| Yes | 236 | 284 | 292 | 440 | 1.28(1.00-1.48) | |

*(Continued)*

**Table 4.** (Continued)

| Subgroups | No COVID-19 vaccination | | COVID-19 vaccination | | aOR(95% CI) a | P for interaction |
|---|---|---|---|---|---|---|
| | Without hay fever | With hay fever | Without hay fever | With hay fever | | |
| **History of COPD** | | | | | | 0.993 |
| No | 6740 | 5708 | 7362 | 7706 | 1.18(1.12-1.24) | |
| Yes | 263 | 468 | 345 | 609 | 1.15(0.88-1.37) | |

Abbreviation: GED, General Educational Development; CVD, Cardiovascular Diseases; BMI; body mass index; aOR,;adjusted odd ratio; Kodachrome Obstructive Pulmonary Disease.

a The analysis included a range of demographic and health-related variables such as age (as a continuous variable), sex (male or female), race/ethnicity (Hispanic, Non-Hispanic White only, Non-Hispanic Black/African American only, Non-Hispanic Asian only, and Non-Hispanic other races/multi-ethnicity), education level (less than high school, high school graduate or GED, and more than high school), cigarette smoking status (nonsmokers, former smokers, and current smokers), urban-rural classification (large central metro, large fringe metro, medium and small metro, and nonmetropolitan), current marital status (married, divorced/separated/widowed, never married, living with a partner, and unknown marital status), region (northeast, midwest, south, and west), family income-to-poverty ratio (below 1, 1.0–2.0, 2.0–3.9, and above 4), BMI categories (underweight, healthy weight, overweight, obese, and unknown), and chronic comorbidities (yes or no for a history of COVID-19, diabetes, hypertension, CVD, chronic obstructive pulmonary disease (COPD), high cholesterol, cancer, anxiety, depression, and a weakened immune system due to prescriptions or health conditions).

b due to prescriptions or health condition.

and depression [17], hypertension and cardiovascular disease [18], 9 and multiple other mental comorbidities [19], many of which are known to increase the risk for atopic disease (i.e., asthma, eczema, hay fever, and food allergy). The comorbid conditions present in atopic disease, use of systemic immunosuppressant medications, increased use of alternative medicines [20] wound lead to the decline of human immune function, increased the allergic reaction of COVID-19 vaccine vaccination, and induce allergic rhinitis, asthma, dermatitis, and allergic skin diseases [21–23]. Previous studies reported that the more COVID-19 vaccination patients accept, the higher their risk of atopic disease and other allergic disorders [24,25]. This could be attributed to higher IL-13 levels in patients with severe COVID19 [26].IL-13 is a critical TH2 cytokine linked to atopic dermatitis exacerbations, which could explain the link between severe COVID-19 infection and worsening of atopic dermatitis control/severity [27]. Hence, COVID-19 infection does not appear to result in worsening atopic dermatitis control/severity in most patients, except those with severe infection [27].

Nevertheless, there is a paucity of data examining the association between COVID-19 vaccination and atopic disease. Our research results show that the correlation between COVID-19 vaccination and hay fever was affected by multiple risk factors. Our study demonstrates higher rates of hay fever in COVID-19 vaccination.

The mechanism of COVID-19 vaccination-induced hay fever is not fully understood yet, but there are several proposed theories [28–34]. One theory suggests that allergic reactions may be due to an immediate hypersensitivity reaction known as Type I IgE-mediated hypersensitivity. In this type of reaction, the body's immune system recognizes a component of the vaccine, such as the mRNA or the lipid nanoparticle used to deliver the mRNA, as a foreign invader. This recognition triggers the release of histamine and other chemicals, leading to allergic symptoms like rash, swelling, and difficulty breathing. Another theory involves the activation of mast cells and basophils. Mast cells are immune cells found in various tissues, including the skin, respiratory system, and digestive tract. Basophils are similar to mast cells and are found in the blood. Both these cells are involved in allergic reactions. When FcεRI on mast cells is cross-linked, it activates the complement system, activates MRGPRX2, causes overstimulation of T cells, and causes monocytes/macrophages to release cytokines, which in turn causes mast cell degranulation as IgE/antigen. In fact, atopic individuals may have greater symptoms than mono-sensitized persons due to their genetic inclination to develop large quantities of antibodies when exposed to several allergens at once [34]. It is suggested that the COVID-19 vaccine may trigger the activation of mast cells and basophils, which leads to the release of histamine and other mediators, resulting in an allergic reaction. Additionally, it is possible that individuals may already have pre-existing allergies that make them more susceptible to developing

**Table 5.** Association of the number of COVID-19 vaccinations with atopic disease among 29 201 participants from the 2021 cycle of the NHIS.

| Atopic Diseases | No COVID-19 Vaccination | 1 vaccination | 2 vaccinations | More than 2 vaccinations | P value for trend [a] |
|---|---|---|---|---|---|
| **Asthma** | | | | | |
| Model 1 [b] | 1[Reference] | 0.91(0.78-1.05) | 1.02(0.95-1.11) | 1.20(1.03–1.40) [c] | 0.041 |
| Model 2 [d] | 1[Reference] | 0.92(0.80-1.07) | 1.06(0.99-1.14) | 1.27(1.09–1.48) [c] | 0.005 |
| Model 3 [e] | 1[Reference] | 0.88(0.76-1.03) | 1.04(0.96-1.12) | 1.21(1.03–1.42) [c] | 0.015 |
| **Skin allergy** | | | | | |
| Model 1 [b] | 1[Reference] | 0.92(0.79-1.08) | 1.04(0.96-1.13) | 1.11(0.94-1.32) | 0.282 |
| Model 2 [d] | 1[Reference] | 0.93(0.79-1.09) | 1.04(0.96-1.12) | 1.10(0.93-1.31) | 0.336 |
| Model 3 [e] | 1[Reference] | 0.89(0.76-1.05) | 1.01(0.93-1.09) | 1.03(0.87-1.22) | 0.508 |
| **Hay fever** | | | | | |
| Model 1 [b] | 1[Reference] | 1.17(1.06–1.23) [c] | 1.25(1.19–1.31) [c] | 1.22(1.37–1.51) [c] | 0.001 |
| Model 2 [d] | 1[Reference] | 1.18(1.07–1.30) [c] | 1.24(1.18–1.30) [c] | 1.17(1.38–1.52) [c] | 0.001 |
| Model 3 [e] | 1[Reference] | 1.15(1.04–1.27) [c] | 1.19(1.13–1.25) [c] | 1.09(0.98–1.22) [c] | 0.001 |
| **Food allergy** | | | | | |
| Model 1 [b] | 1[Reference] | 1.19(1.02–1.39) [c] | 1.04(0.96-1.13) | 1.03(0.86-1.24) | 0.153 |
| Model 2 [d] | 1[Reference] | 1.18(1.01–1.38) [c] | 1.01(0.93-1.10) | 0.98(0.82-1.18) | 0.19 |
| Model 3 [e] | 1[Reference] | 1.16(0.99-1.36) | 0.99(0.91-1.08) | 0.94(0.78-1.13) | 0.21 |

Abbreviations: NHIS, National Health Interview Survey; OR, odd ratio.

a P < .05 was considered statistically significant.

b Model 1 was adjusted for age (continuous), and sex (male or female).

c Statistically significant.

d Model 2 incorporated all the variables from Model 1 and further adjusted for race/ethnicity (Hispanic, Non-Hispanic White only, Non-Hispanic Black/African American only, Non-Hispanic Asian only, Non-Hispanic other races, and other/multi-ethnicity), education level (less than high school, high school graduate or GED, and more than high school), cigarette smoking status (nonsmokers, former smokers, and current smokers), urban-rural classification (large central metro, large fringe metro, medium and small metro, and nonmetropolitan), current marital status (married, divorced/separated/widowed, never married, living with a partner, and unknown marital status), region (northeast, midwest, south, and west), and family income-to-poverty ratio (below 1, 1.0–1.9, 2.0–3.9, and above 4).

e Model 3 was adjusted for the variables in model 2 plus BMI categories (underweight, healthy weight, overweight, obese, and unknown), and chronic comorbidities [ever had COVID-19,history of diabetes, history of hypertension, history of CVD, history of COPD, history of high cholesterol level, history of cancer, History of anxiety, history of depression, and history of weakened immune system(due to prescriptions or health condition)].

an allergic reaction to the vaccine. This could be due to sensitization to certain components of the vaccine or previous allergies to other medications or substances.

On the other hand, our sensitivity analysis showed that the association between COVID-19 vaccine and the risk of hay fever and asthma increased with the increase of the number of COVID-19 vaccines, while skin allergy and food allergy were not. Previous reports suggested that patients may become IgE-sensitized by previous exposure to antigens [34], therefore atopic patients could be more vulnerable in the second dose of vaccination. However, our studies had shown that, compared with individuals without COVID-19 vaccination groups, COVID-19 vaccination groups have not exacerbated of chronic immune- mediated dermatoses like skin allergy and food allergy. This may be due to COVID-19 vaccines are non-live vaccines. Despite the fact that individuals with atopic illness are known to be at a higher risk of egg-related allergic responses [35], COVID-19 vaccinations are not egg-based and do not include this risk.

## Public health implication

With the end of the Public Health Emergency, COVID-19 surveillance in the US is no longer performed using case counts. People slowly forgot about the new coronavirus, but it did not disappear, the new coronavirus and its variants are still

constantly circulating among countries around the world. However, a recently reported that older adults made up more than half of COVID-19 hospitalizations in the US from October 2023 to December 2023 [36]. Vaccination is still an important part of the current global effort to combat the COVID-19 pandemic. Other studies have reported that the electronic health record data collected from the whole population aged 5 years or older showed that compared with the people who were fully vaccinated, the risk of hospitalization or death due to COVID-19 among British people who did not receive all eligible vaccines was up to four times higher [37]. Although our resulted indicated that vaccinated individuals were about 1.2 times as likely as unvaccinated individuals to develop hay fever and asthma in the absence of COVID-19 infection, this increased risk must be weighed against the overall benefits of vaccination. The decision-making on risk management in public health practice must take into account many complex and potentially contradictory factors [38]. When trying to balance risk and benefits, one should consider the types of advantages and the people who gain them, especially when there are offsetting risks or unavoidable benefits.

## Strengths and limitations

This study has several strengths. Compared with previous studies on COVID-19 vaccinations and one or two atopic diseases, our study included a large, representative sample of US adults and four atopic diseases. Moreover, we found that COVID-19 vaccination and vaccination dose are correlated with the occurrence of hay fever; There was no correlation with the occurrence of food allergy and skin allergy. However, with the increase of vaccination dose, the prevalence of hay fever and asthma attacks will gradually increase. Our study has several limitations. First, because the survey was cross-sectional, cause-effect inferences cannot be made. Second, even after accounting for several potential confounders at the beginning, residual confounding—like physical activity, alcohol consumption, and dietary factors—cannot be completely ruled out. Third, the medical history of atopic diseases was self-reported, which may miss the estimation of atopic diseases and had not been validated through diagnostic testing; COVID-19 vaccination cannot be directly attributed to skin-specific. When data are available, future studies using the NHIS Adult Study may directly assess adult vaccination against COVID-19. Fourth, NIHS only includes noninstitutionalized civilians; thus, this analysis may underestimate certain parts of the US population. Fifth, this study was not possible to eliminate those who had a history of medication allergies or other allergic illnesses. However, prior research shown that, in individuals with a history of medication allergies and other allergic disorders, allergic responses to COVID-19 vaccinations are uncommon and, for the most part, minor and readily manageable [24]. A possible limitation should be noted that vaccinated individuals may have had more healthcare interactions, leading to higher likelihood of detection bias. Additionally, the 6-month window used in this study to identify new diagnoses of atopic diseases is relatively short compared to the potential chronic nature of these conditions. Chronic atopic disease often has a protracted course of development and may not manifest immediately following vaccination. The study's reliance on a short follow-up period may therefore fail to capture the full impact of COVID-19 vaccination on the incidence of these conditions. Finally, the prospective of the study precludes the establishment of a temporal relationship between vaccination and the onset of atopic diseases. Future research should employ longitudinal study designs to better understand the long-term associations between COVID-19 vaccination and the development of atopic diseases.

## Conclusions

In 2021 nationally representative survey of US adults, COVID-19 vaccination was associated with short-term higher risk incidence of hay fever, but COVID-19 vaccination was not associated with an increased in incidence risk in skin allergy, asthma and food allergy.

## Acknowledgments

We appreciate all participants in the NHANES.

## Author contributions

**Conceptualization:** Min Lu, Zixuan Bu.

**Data curation:** Min Lu, Zixuan Bu, Nana Xiang, juebo yu.

**Formal analysis:** Min Lu, juebo yu.

**Funding acquisition:** Min Lu, Zixuan Bu.

**Investigation:** Min Lu, juebo yu.

**Methodology:** Min Lu, Nana Xiang, juebo yu.

**Project administration:** juebo yu.

**Resources:** Nana Xiang.

**Software:** Min Lu, juebo yu.

**Supervision:** juebo yu.

**Validation:** Min Lu, Nana Xiang.

**Visualization:** juebo yu.

**Writing – original draft:** juebo yu.

**Writing – review & editing:** Min Lu, Zixuan Bu, Nana Xiang, juebo yu.

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
