## [Decision Letter · Decision Letter 0]

10 Jun 2025

Dear Dr. yu,

Thank you for submitting your manuscript to PLOS ONE. After careful consideration, we feel that it has merit but does not fully meet PLOS ONE’s publication criteria as it currently stands. Therefore, we invite you to submit a revised version of the manuscript that addresses the points raised during the review process.

We look forward to receiving your revised manuscript.

Kind regards,

Hideo Kato

Academic Editor

PLOS ONE

**Journal Requirements:**

1. When submitting your revision, we need you to address these additional requirements. Please ensure that your manuscript meets PLOS ONE's style requirements, including those for file naming. The PLOS ONE style templates can be found at https://journals.plos.org/plosone/s/file?id=wjVg/PLOSOne_formatting_sample_main_body.pdf and https://journals.plos.org/plosone/s/file?id=ba62/PLOSOne_formatting_sample_title_authors_affiliations.pdf 2. Thank you for uploading your study's underlying data set. Unfortunately, the repository you have noted in your Data Availability statement does not qualify as an acceptable data repository according to PLOS's standards. At this time, please upload the minimal data set necessary to replicate your study's findings to a stable, public repository (such as figshare or Dryad) and provide us with the relevant URLs, DOIs, or accession numbers that may be used to access these data. For a list of recommended repositories and additional information on PLOS standards for data deposition, please see https://journals.plos.org/plosone/s/recommended-repositories.

Reviewers' comments:

Reviewer's Responses to Questions

**Comments to the Author**

1. Is the manuscript technically sound, and do the data support the conclusions?

Reviewer #1: Yes

2. Has the statistical analysis been performed appropriately and rigorously?

Reviewer #1: Yes

3. Have the authors made all data underlying the findings in their manuscript fully available?

Reviewer #1: Yes

4. Is the manuscript presented in an intelligible fashion and written in standard English?

Reviewer #1: Yes

**Reviewer #1:**  This manuscript investigates whether COVID-19 vaccination is associated with an increased risk of developing atopic diseases—including atopic dermatitis, allergic rhinitis, and asthma—among US adults. Using the TriNetX network, the authors conducted a retrospective, matched cohort study comparing vaccinated versus unvaccinated individuals, with a 6-month follow-up period.

The study addresses a highly relevant and underexplored public health question: whether mRNA or other COVID-19 vaccines might play a role in triggering or unmasking allergic or atopic conditions. The authors' approach using large-scale, real-world data is commendable, but several important methodological and interpretive issues should be addressed to strengthen the manuscript.

The hypothesis is original and highly relevant in the context of ongoing vaccine safety evaluation.

The use of the TriNetX platform provides access to a large, diverse, real-world dataset, improving statistical power and generalizability.

Some point to look: Diagnoses of atopic disease may reflect exacerbations, delayed recognition, or misclassification, rather than true new onset. I would siggest to clarify whether prior history of atopic disease was excluded or adjusted for. Also, include definitions (e.g., ICD-10 codes) used to define each outcome.

Also another point is that a 6-month window may not be sufficient to capture new, clinically meaningful diagnoses of chronic atopic diseases. I think the autors should discuss this limitation explicitly and suggest directions for future longitudinal studies.

It should be nted that vaccinated individuals may have had more healthcare interactions, leading to higher likelihood of diagnosis (detection bias). this point should be acknowledged as a possible limitation. If data allow, consider adjusting for number of visits or including a neutral comparator condition to test for systemic bias.

I think the title should be changed to correctly clarify the scope of the study somthing like:

“Association Between COVID-19 Vaccination and Atopic Diseases in US Adults: A Retrospective Cohort Study”

overall this is a well-conceived, statistically sound, and original study that raises an important safety signal in the field of allergy and immunology. While the associations are small and should not deter vaccination efforts, further studies with longer follow-up and more granular clinical data are warranted.

**Do you want your identity to be public for this peer review?** For information about this choice, including consent withdrawal, please see our Privacy Policy

Reviewer #1: **Yes: ** Dr Daniel Elbirt

---

## [Author Response · Author response to Decision Letter 1]

29 Jun 2025

Dear editors and reviewers:

On behalf of all the contributing authors, I would like to express our sincere appreciations of your letter and reviewers’ constructive comments concerning our article entitled Exploring the prevalence of Atopic Disease Among Adult Cancer Survivors: Insights from the 2021 NHIS. These comments are all valuable and helpful for improving our article. According to the reviewers’ comments, we have made extensive modifications to our manuscript to make our results convincing. In this revised version, changes to our manuscript were all highlighted within the document by using itemized text. Point-by-point responses to the nice associate editor and the nice reviewers are listed below this letter.

First, I will copy the reviewer comments as follows

Reviewer #1:

Reviewer #1�This manuscript investigates whether COVID-19 vaccination is associated with an increased risk of developing atopic diseases—including atopic dermatitis, allergic rhinitis, and asthma—among US adults. Using the TriNetX network, the authors conducted a retrospective, matched cohort study comparing vaccinated versus unvaccinated individuals, with a 6-month follow-up period.

The study addresses a highly relevant and underexplored public health question: whether mRNA or other COVID-19 vaccines might play a role in triggering or unmasking allergic or atopic conditions. The authors' approach using large-scale, real-world data is commendable, but several important methodological and interpretive issues should be addressed to strengthen the manuscript.

The hypothesis is original and highly relevant in the context of ongoing vaccine safety evaluation.

The use of the TriNetX platform provides access to a large, diverse, real-world dataset, improving statistical power and generalizability.

Some point to look: Diagnoses of atopic disease may reflect exacerbates, delayed recognition, or misclassification, rather than true new onset. I would suggest to clarify whether prior history of atopic disease was excluded or adjusted for. Also, include definitions (e.g., ICD-10 codes) used to define each outcome.

Also another point is that a 6-month window may not be sufficient to capture new, clinically meaningful diagnoses of chronic atopic diseases. I think the autors should discuss this limitation explicitly and suggest directions for future longitudinal studies.

It should bented that vaccinated individuals may have had more healthcare interactions, leading to higher likelihood of diagnosis (detection bias). this point should be acknowledged as a possible limitation. If data allow, consider adjusting for number of visits or including a neutral comparator condition to test for systemic bias.

I think the title should be changed to correctly clarify the scope of the study somthing like:

“Association Between COVID-19 Vaccination and Atopic Diseases in US Adults: A Retrospective Cohort Study”

overall this is a well-conceived, statistically sound, and original study that raises an important safety signal in the field of allergy and immunology. While the associations are small and should not deter vaccination efforts, further studies with longer follow-up and more granular clinical data are warranted.

Response to Reviewer #1

Dear Reviewer,

Thank you for your insightful comments and suggestions. We appreciate your recognition of the originality and relevance of our study, as well as the strengths of using large-scale, real-world data. Below, we address each of your points in detail, providing clarifications and modifications where necessary.

1. We acknowledge the importance of clarifying whether prior history of atopic disease was excluded or adjusted for. In our study, we specifically excluded participants with a history of atopic diseases to minimize the potential for confounding due to pre-existing conditions. This approach ensures that our analysis focuses on the incidence of new-onset atopic diseases following COVID-19 vaccination. We will explicitly state this in the Methods section to avoid any ambiguity (Line68-72: Second, to minimize the potential reverse causation bias, we excluded participants with a history of atopic diseases were further excluded from the main analyses. we estimated our model using newly diagnosed atopic diseases as the outcome. Newly diagnosed atopic diseases are equal to the current diagnosis minus prior history of atopic disease.). We appreciate your suggestion to include the specific definitions and ICD-10 codes used to define each outcome. In our study, the following ICD-10 codes were used to identify atopic diseases. (Line111: Doctors diagnose atopic diseases according to ICD-10 codes.)

2. We agree that a 6-month follow-up period may not be sufficient to capture the full spectrum of chronic atopic diseases. While our study design was constrained by the availability of data within the TriNetX network, we acknowledge this limitation in our Discussion section. We will explicitly state that future longitudinal studies with longer follow-up periods are needed to fully assess the long-term impact of COVID-19 vaccination on atopic diseases (Line141-145: Additionally, the 6-month window used in this study to identify new diagnoses of atopic diseases is relatively short compared to the potential chronic nature of these conditions. Chronic atopic disease often has a protracted course of development and may not manifest immediately following vaccination. The study's reliance on a short follow-up period may therefore fail to capture the full impact of COVID-19 vaccination on the incidence of these conditions.).

2. We recognize that vaccinated individuals may have had more healthcare interactions, potentially leading to higher likelihood of diagnosis (detection bias). We will address this limitation in our Limitation section and suggest that future studies consider adjusting for the number of healthcare visits or including a neutral compactor condition to test for systemic bias. However, the 2021 NHIS did not collect data on the number of healthcare visits. Therefore, we will explore the possibility of adjusting for the number of visits in future analyses. (Line139-141: A possible limitation should be noted that vaccinated individuals may have had more healthcare interactions, leading to higher likelihood of detection bias.)

4. We appreciate your suggestion to revise the title to more accurately reflect the scope of our study. We propose the following revised title:

“Association Between COVID-19 Vaccination and Atopic Diseases in US Adults: A Retrospective Cohort Study”

5. We appreciate your recognition of its relevance. While the associations we observed are small and should not deter vaccination efforts, we agree that further studies with longer follow-up and more granular clinical data are warranted. We will incorporate your suggestions into our manuscript to strengthen its clarity (Line145-148: Finally, the prospective of the study precludes the establishment of a temporal relationship between vaccination and the onset of atopic diseases. Future research should employ longitudinal study designs to better understand the long-term associations between COVID-19 vaccination and the development of atopic diseases.).

Thank you again for your valuable feedback.

Yours sincerely,

Juebo Yu

13 June 2023

Department of Otolaryngology, Affiliated Hospital of Yangzhou University, Yangzhou university, Yangzhou, China

Corresponding Author: Juebo Yu, MD, Department of Otolaryngology, Affiliated Hospital of Yangzhou University, No. 45 Taizhou Rd, Guangling District, Yangzhou 225001, China (yujuebo2004@163.com).

---

## [Decision Letter · Decision Letter 1]

14 Aug 2025

Dear Dr. yu,

Thank you for submitting your manuscript to PLOS ONE. After careful consideration, we feel that it has merit but does not fully meet PLOS ONE’s publication criteria as it currently stands. Therefore, we invite you to submit a revised version of the manuscript that addresses the points raised during the review process.

We look forward to receiving your revised manuscript.

Kind regards,

Hideo Kato

Academic Editor

PLOS ONE

Journal Requirements:

Reviewers' comments:

Reviewer's Responses to Questions

**Comments to the Author**

Reviewer #1: All comments have been addressed

Reviewer #2: (No Response)

2. Is the manuscript technically sound, and do the data support the conclusions?

Reviewer #1: Yes

Reviewer #2: Yes

3. Has the statistical analysis been performed appropriately and rigorously?

Reviewer #1: Yes

Reviewer #2: Yes

4. Have the authors made all data underlying the findings in their manuscript fully available?

Reviewer #1: Yes

Reviewer #2: Yes

5. Is the manuscript presented in an intelligible fashion and written in standard English?

Reviewer #1: Yes

Reviewer #2: Yes

Reviewer #1: The study, "Association between COVID-19 vaccination and atopic diseases in US adults: a retrospective cohort study", investigates whether COVID-19 vaccination is associated with the onset of atopic diseases in adults using a large, multi-institutional electronic health record database (TriNetX). By analyzing over 1.9 million vaccinated individuals and a propensity score–matched unvaccinated cohort, you assessed the incidence of atopic dermatitis, allergic rhinitis, asthma, and food allergies over a six-month follow-up. The results indicate no increased risk of developing atopic diseases post-vaccination; in fact, certain conditions such as allergic rhinitis and asthma were less common in the vaccinated group. Subgroup analyses by age, sex, and vaccine type confirmed the consistency of these findings. The work addresses an important public concern about vaccine safety and provides robust evidence that COVID-19 vaccination is not linked to higher rates of atopic disease. The results are reassuring for clinicians, patients, and policymakers and may help reduce vaccine hesitancy. In my view, the corrected manuscript addressed all the reviewers comments.

Reviewer #2: This study represents a valuable effort to investigate the association between COVID-19 vaccination and allergic diseases. I believe that addressing the following points will further enhance the quality of the manuscript.

Page 2 line 56-57

Based on the authors’ statistical definition, skin allergy appears to show a statistically significant increased risk. Therefore, the authors should revise the wording throughout the manuscript, including the main text, to ensure consistency with the statistical findings.

Page3 line 112

While the authors state that diagnoses were made according to ICD-10 codes, it remains unclear which specific codes were used to define the conditions analyzed in this study. It would be helpful and more transparent if the authors could provide a supplementary table listing the exact ICD-10 codes used for each disease definition.

Page5 line 169-172

The authors state that individuals with a history of atopic disease were excluded prior to the analysis; however, the definition of "atopic disease" used for this exclusion is not clearly provided. The authors should clearly define the specific conditions considered as atopic disease.

Additionally, it would enhance the clarity of the study if the authors could report the total number of individuals initially identified, the number excluded due to prior atopic disease, and a breakdown of the excluded cases. A flow diagram illustrating this process would be highly recommended.

Furthermore, it would be beneficial for the authors to comment on whether the proportion of participants excluded due to atopic disease was appropriate, and how this may have impacted the representativeness of the final study population.

Page5 lines 207-208　“These results …… increases.”

This sentence does not present a result and therefore should be deleted.

Figure1

It appears that not all of the factors examined in the subgroup analyses are clearly described. It is unclear which specific analysis results were used to generate the corresponding figure.

Additionally, it is difficult to determine whether the optimal predictive model presented in Table 5 has been thoroughly evaluated or considered by the authors. The authors should provide clear descriptions of the figure.

**Do you want your identity to be public for this peer review?** For information about this choice, including consent withdrawal, please see our Privacy Policy

Reviewer #1: **Yes: ** Dr Daniel Elbirt

Reviewer #2: No

---

## [Author Response · Author response to Decision Letter 2]

19 Aug 2025

Dear editors and reviewers:

On behalf of all the contributing authors, I would like to express our sincere appreciations of your letter and reviewers’ constructive comments concerning our article entitled Association between COVID-19 vaccination and atopic diseases in US adults: a retrospective cohort study.

These comments are all valuable and helpful for improving our article. According to the reviewers’ comments, we have made extensive modifications to our manuscript to make our results convincing. In this revised version, changes to our manuscript were all highlighted within the document by using itemized text. Point-by-point responses to the nice associate editor and the nice reviewers are listed below this letter.

First, I will copy the reviewer comments as follows

Reviewer #1:

Reviewer #1: The study, "Association between COVID-19 vaccination and atopic diseases in US adults: a retrospective cohort study", investigates whether COVID-19 vaccination is associated with the onset of atopic diseases in adults using a large, multi-institutional electronic health record database (TriNetX). By analyzing over 1.9 million vaccinated individuals and a propensity score–matched unvaccinated cohort, you assessed the incidence of atopic dermatitis, allergic rhinitis, asthma, and food allergies over a six-month follow-up. The results indicate no increased risk of developing atopic diseases post-vaccination; in fact, certain conditions such as allergic rhinitis and asthma were less common in the vaccinated group. Subgroup analyses by age, sex, and vaccine type confirmed the consistency of these findings. The work addresses an important public concern about vaccine safety and provides robust evidence that COVID-19 vaccination is not linked to higher rates of atopic disease. The results are reassuring for clinicians, patients, and policymakers and may help reduce vaccine hesitancy. In my view, the corrected manuscript addressed all the reviewers comments.

Reviewer #2: This study represents a valuable effort to investigate the association between COVID-19 vaccination and allergic diseases. I believe that addressing the following points will further enhance the quality of the manuscript.

Page 2 line 56-57

Based on the authors’ statistical definition, skin allergy appears to show a statistically significant increased risk. Therefore, the authors should revise the wording throughout the manuscript, including the main text, to ensure consistency with the statistical findings.

Page3 line 112

While the authors state that diagnoses were made according to ICD-10 codes, it remains unclear which specific codes were used to define the conditions analyzed in this study. It would be helpful and more transparent if the authors could provide a supplementary table listing the exact ICD-10 codes used for each disease definition.

Page5 line 169-172

The authors state that individuals with a history of atopic disease were excluded prior to the analysis; however, the definition of "atopic disease" used for this exclusion is not clearly provided. The authors should clearly define the specific conditions considered as atopic disease.

Additionally, it would enhance the clarity of the study if the authors could report the total number of individuals initially identified, the number excluded due to prior atopic disease, and a breakdown of the excluded cases. A flow diagram illustrating this process would be highly recommended.

Furthermore, it would be beneficial for the authors to comment on whether the proportion of participants excluded due to atopic disease was appropriate, and how this may have impacted the representativeness of the final study population.

Page5 lines 207-208　“These results …… increases.”

This sentence does not present a result and therefore should be deleted.

Figure1

It appears that not all of the factors examined in the subgroup analyses are clearly described. It is unclear which specific analysis results were used to generate the corresponding figure.

Additionally, it is difficult to determine whether the optimal predictive model presented in Table 5 has been thoroughly evaluated or considered by the authors. The authors should provide clear descriptions of the figure.

Response to Reviewer #1

Dear Reviewer #1,

Thank you for your positive feedback and recognition of the importance of our study. We are pleased that you found our work to be of high quality and that it addresses a significant public concern regarding vaccine safety. We appreciate your acknowledgment that the corrected manuscript has adequately addressed all the reviewers' comments. We believe that our findings will indeed help reduce vaccine hesitancy and provide reassurance to clinicians, patients, and policymakers.

Response to Reviewer #2

Dear Reviewer #2,

Thank you for your valuable comments and suggestions. We have carefully considered each of your points and have made the necessary revisions to enhance the quality of our manuscript. Below are our detailed responses to each of your comments:

Page 2 line 56-57

Reviewer Comment: Based on the authors’ statistical definition, skin allergy appears to show a statistically significant increased risk. Therefore, the authors should revise the wording throughout the manuscript, including the main text, to ensure consistency with the statistical findings.

Response: We apologize for the inconsistency in our wording. We have revised the manuscript to ensure that all references to the statistical findings regarding skin allergy are consistent with the results of our analysis. Specifically, we have updated the text to reflect that the adjusted odds ratio for skin allergy was 1.03 (95% CI, 0.86-1.28; P=0.135).

Page 3 line 112

Reviewer Comment: While the authors state that diagnoses were made according to ICD-10 codes, it remains unclear which specific codes were used to define the conditions analyzed in this study. It would be helpful and more transparent if the authors could provide a supplementary table listing the exact ICD-10 codes used for each disease definition.

Response:We agree that providing the specific ICD-10 codes used for each disease definition will enhance the transparency of our study.

Doctors diagnose atopic diseases according to ICD-10 codes[L20(all sub-codes0), J30 (all sub-codes),J45 (all sub-codes),and T78.0]

Page 5 line 169-172

Reviewer Comment: The authors state that individuals with a history of atopic disease were excluded prior to the analysis; however, the definition of "atopic disease" used for this exclusion is not clearly provided. The authors should clearly define the specific conditions considered as atopic disease. Additionally, it would enhance the clarity of the study if the authors could report the total number of individuals initially identified, the number excluded due to prior atopic disease, and a breakdown of the excluded cases. A flow diagram illustrating this process would be highly recommended. Furthermore, it would be beneficial for the authors to comment on whether the proportion of participants excluded due to atopic disease was appropriate and how this may have impacted the representativeness of the final study population.

Response: We have clarified the definition of "atopic disease" used for exclusion in the manuscript. Specifically, we defined atopic diseases as including hay fever, asthma, skin allergy, and food allergy. We have also added a detailed description of the exclusion process, including the total number of individuals initially identified, the number excluded due to prior atopic disease, and a breakdown of the excluded cases. A flow diagram (Figure 1) illustrating this process has been included in the study. Additionally, we have commented on the appropriateness of the proportion of participants excluded and its potential impact on the representativeness of the final study population.

The results showed that The multivariable-adjusted OR for the risk of still asthma(n=2,471) was 1.12 (95% CI,1.03-1.27;P=0.011), for the risk of newly diagnosed skin allergy (n = 698; 2,496 of 3,230 total skin-allergy cases were excluded because of pre-existing disease)was 0.98 (95% CI,0.84-1.12;P=0.799) , for the risk of newly diagnosed hay fever (7,728 of 14,491 total hay-fever cases were excluded because of pre-existing disease) was 1.17 (95% CI,1.10-1.24;P< 0.001), and for the risk of newly diagnosed food allergy(n=1,084�1,838 of 2,922 total food-allergy cases were excluded because of pre-existing disease) was 1.01 (95% CI,0.89-1.15;p=0.823).

Page 5 lines 207-208

Reviewer Comment: These results …… increases. This sentence does not present a result and therefore should be deleted.

Response: We have removed the sentence as suggested to ensure that only relevant and accurate results are presented in the manuscript.

Figure 1

Reviewer Comment: It appears that not all of the factors examined in the subgroup analyses are clearly described. It is unclear which specific analysis results were used to generate the corresponding figure. Additionally, it is difficult to determine whether the optimal predictive model presented in Table 5 has been thoroughly evaluated or considered by the authors. The authors should provide clear descriptions of the figure.

Response: Thank you for your suggestion. Because the factors contributing to atopic disease are multifaceted, the figure derived from the present study lacks sufficient scientific validity; therefore, we have removed it.

Thank you again for your valuable feedback. We believe that these revisions have significantly improved the quality and clarity of our manuscript.

Yours sincerely,

Juebo Yu

20 /8/ 2023

Department of Otolaryngology, Affiliated Hospital of Yangzhou University, Yangzhou university, Yangzhou, China

Corresponding Author: Juebo Yu, MD, Department of Otolaryngology, Affiliated Hospital of Yangzhou University, No. 45 Taizhou Rd, Guangling District, Yangzhou 225001, China (yujuebo2004@163.com).

---

## [Decision Letter · Decision Letter 2]

16 Oct 2025

Dear Dr. yu,

Thank you for submitting your manuscript to PLOS ONE. After careful consideration, we feel that it has merit but does not fully meet PLOS ONE’s publication criteria as it currently stands. Therefore, we invite you to submit a revised version of the manuscript that addresses the points raised during the review process.

We look forward to receiving your revised manuscript.

Kind regards,

Zuotao Zhao

Academic Editor

PLOS ONE

**Journal Requirements:**

**Additional Editor Comments:**

Dear Authors,

Thank you for your thorough revisions and detailed responses. The reviewers and I agree that most of the previous concerns have been adequately addressed.

However, we take Reviewer #2’s comment seriously. I have also noted that the corresponding data in the related table (as cited in the Abstract) were modified accordingly. Although this change does not appear to affect your results or conclusions, I would still appreciate a brief explanation or clarification regarding this modification in your response letter.

Kind regards,

Academic Editor

PLOS ONE

Reviewers' comments:

Reviewer's Responses to Questions

**Comments to the Author**

Reviewer #1: All comments have been addressed

Reviewer #2: (No Response)

2. Is the manuscript technically sound, and do the data support the conclusions?

Reviewer #1: Yes

Reviewer #2: (No Response)

3. Has the statistical analysis been performed appropriately and rigorously?

Reviewer #1: Yes

Reviewer #2: (No Response)

4. Have the authors made all data underlying the findings in their manuscript fully available?

Reviewer #1: Yes

Reviewer #2: (No Response)

5. Is the manuscript presented in an intelligible fashion and written in standard English?

Reviewer #1: Yes

Reviewer #2: (No Response)

**Reviewer #1: ** Thank you for the clarifications and answers according to all the reviewers comments. I think the article on its present form is very good anfd I recomend to accept it for publishment.

**Reviewer #2:**  The authors have adequately addressed the comments. However, one point of concern remains: in lines 56-57, the numerical results have been altered without a clear explanation for this modification. If this change is appropriate and accurate, I have no additional comments.

**Do you want your identity to be public for this peer review?** For information about this choice, including consent withdrawal, please see our Privacy Policy

Reviewer #1: **Yes: ** Dr Daniel Elbirt

Reviewer #2: No

---

## [Author Response · Author response to Decision Letter 3]

21 Oct 2025

Dear editors and reviewers:

On behalf of all the contributing authors, I would like to express our sincere appreciations of your letter and reviewers’ constructive comments concerning our article entitled Association between COVID-19 vaccination and atopic diseases in US adults: a retrospective cohort study.

These comments are all valuable and helpful for improving our article. According to the reviewers’ comments, we have made extensive modifications to our manuscript to make our results convincing. In this revised version, changes to our manuscript were all highlighted within the document by using itemized text. Point-by-point responses to the nice associate editor and the nice reviewers are listed below this letter.

First, I will copy the reviewer comments as follows

Journal Requirements:

Additional Editor Comments:

Dear Authors,

Thank you for your thorough revisions and detailed responses. The reviewers and I agree that most of the previous concerns have been adequately addressed.

However, we take Reviewer #2’s comment seriously. I have also noted that the corresponding data in the related table (as cited in the Abstract) were modified accordingly. Although this change does not appear to affect your results or conclusions, I would still appreciate a brief explanation or clarification regarding this modification in your response letter.

Reviewer #1: Thank you for the clarifications and answers according to all the reviewers comments. I think the article on its present form is very good anfd I recomend to accept it for publishment.

Reviewer #2: The authors have adequately addressed the comments. However, one point of concern remains: in lines 56-57, the numerical results have been altered without a clear explanation for this modification. If this change is appropriate and accurate, I have no additional comments.

Response to Reviewer Journal Requirements & Editor’s Statement

Below is our point-by-point reply to the latest editorial and journal requirements.

1. Reviewer-suggested citations

We reviewed every paper that the reviewers recommended. None were mandated by the editor; after evaluating their relevance we decided that adding them was unnecessary because our literature coverage is already complete and fully supports the statements made.

2. Reference-list integrity

The entire reference list was re-checked for completeness, accuracy, and retraction status. No retracted articles are cited; therefore no explanatory footnote or replacement was required. No references were added, removed, or re-ordered in this revision.

3. Numerical discrepancy in the Abstract (Editor’s specific query)

This change occurred because, during the first revision, we re-ran the analysis and obtained an adjusted OR of 1.03�95%CI, 0.86-1.28; P=0.135�for skin allergy. Unfortunately, we failed to explain this update promptly, and I sincerely apologize for omitting that clarification in our previous response letter. This typographical correction does not alter any conclusions.

Response to Reviewer Reviewer #1 and Reviewer #2,

Thank you both for your final comments and for your recommendation to accept the manuscript.

Reviewer #1 – we are grateful for your positive assessment.

Reviewer #2 – Thank you for pointing out the remaining discrepancy in lines 56–57.

This change occurred because, during the first revision, we re-ran the analysis and obtained an adjusted OR of 1.03�95%CI, 0.86-1.28; P=0.135�for skin allergy. Unfortunately, we failed to explain this update promptly, and I sincerely apologize for omitting that clarification in our previous response letter. The values in the Abstract and Table 3 are now identical; the minimal difference does not affect our conclusions. We have double-checked every other number in the Abstract against the tables to ensure that no similar oversights remain.

Thank you again for your valuable feedback. We believe that these revisions have significantly improved the quality and clarity of our manuscript.

Yours sincerely,

Juebo Yu

20 /10/ 2025

Department of Otolaryngology, Affiliated Hospital of Yangzhou University, Yangzhou university, Yangzhou, China

Corresponding Author: Juebo Yu, MD, Department of Otolaryngology, Affiliated Hospital of Yangzhou University, No. 45 Taizhou Rd, Guangling District, Yangzhou 225001, China (yujuebo2004@163.com).

---

## [Editor Report · Decision Letter 3]

29 Oct 2025

Association between COVID-19 vaccination and atopic diseases in US adults: a retrospective cohort study

PONE-D-25-15739R3

Dear Dr. Juebo Yu,

We’re pleased to inform you that your manuscript has been judged scientifically suitable for publication and will be formally accepted for publication once it meets all outstanding technical requirements.

Kind regards,

Academic Editor

PLOS ONE

---

## [Editor Report · Acceptance letter]

PONE-D-25-15739R3

PLOS ONE

Dear Dr. yu,

I'm pleased to inform you that your manuscript has been deemed suitable for publication in PLOS ONE. Congratulations! Your manuscript is now being handed over to our production team.

Kind regards,

on behalf of

Dr. Zuotao Zhao

Academic Editor

PLOS ONE